# Rapid Depletions of Subcutaneous Fat Mass and Skeletal Muscle Mass Predict Worse Survival in Patients with Hepatocellular Carcinoma Treated with Sorafenib

**DOI:** 10.3390/cancers11081206

**Published:** 2019-08-19

**Authors:** Kenji Imai, Koji Takai, Takao Miwa, Daisuke Taguchi, Tatsunori Hanai, Atsushi Suetsugu, Makoto Shiraki, Masahito Shimizu

**Affiliations:** Department of Gastroenterology/Internal Medicine, Gifu University Graduate School of Medicine, 1-1 Yanagido, Gifu 501-1194, Japan

**Keywords:** hepatocellular carcinoma, skeletal muscle, subcutaneous fat, sarcopenia, cancer cachexia, sorafenib

## Abstract

The aim of this study was to investigate whether rapid depletions of fat mass and skeletal muscle mass predict mortality in hepatocellular carcinoma (HCC) patients treated with sorafenib. This retrospective study evaluated 61 HCC patients. The cross-sectional areas of visceral and subcutaneous fat mass and skeletal muscle mass were measured by computed tomography, from which the visceral fat mass index (VFMI), subcutaneous fat mass index (SFMI), and skeletal muscle index (L3SMI) were obtained. The relative changes in these indices per 120 days (ΔVFMI, ΔSFMI, and ΔL3SMI) before and after sorafenib treatment were calculated in each patient. Patients within the 20th percentile cutoffs for these indices were classified into the rapid depletion (RD) group. Kaplan–Meier analysis revealed that with respect to ΔL3SMI (*p* = 0.0101) and ΔSFMI (*p* = 0.0027), the RD group had a significantly poorer survival. Multivariate analysis using the Cox proportional-hazards model also demonstrated that ΔL3SMI (≤−5.73 vs. >−5.73; hazard ratio [HR]: 4.010, 95% confidence interval [CI]: 1.799–8.938, *p* = < 0.001) and ΔSFMI (≤−5.33 vs. >−5.33; HR: 4.109, 95% CI: 1.967–8.584, *p* = < 0.001) were independent predictors. Rapid depletions of subcutaneous fat mass and skeletal muscle mass after the introduction of sorafenib indicate a poor prognosis.

## 1. Introduction

Hepatocellular carcinoma (HCC) is one of the most common malignancies worldwide. Yearly, more than half a million people are diagnosed with HCC worldwide [1]. The prognosis of HCC is notably poor because only 46% of the patients are diagnosed at an early stage, and most do not receive curative therapy [2,3]. Furthermore, even those who undergo curative therapy are usually prone to recurrence because HCC generally develops in patients with liver cirrhosis, which is regarded as a hyper-carcinogenic disease [4]. In fact, the 5-year recurrence rate after curative treatment is more than 70% [5,6]. Therefore, in addition to early detection and radical treatment for initial HCC, adequate therapy is critical to improve the prognosis of patients with this malignancy even when they are at an advanced stage.

Sorafenib is the first orally active multi-kinase inhibitor that has been confirmed to be efficacious against advanced HCC [7]. Sorafenib is first recommended in cases with Child-Pugh’s A or B class and advanced stage [8,9,10], and is widely used currently to treat advanced HCC. The prognosis of patients with advanced HCC is significantly improved by sorafenib treatment [7,11]. However, several adverse events including hand-foot syndrome, fatigue, and rash can occur when using this agent. Digestive symptoms such as anorexia, nausea, and vomiting are also frequently observed in sorafenib-treated patients [7,11]. These adverse events cause malnutrition and unfavorable changes in body composition, which are involved in the worse survival of HCC patients [12]. Therefore, it is important to adequately deal with various nutritional challenges that arise during sorafenib treatment.

Skeletal muscle depletion or sarcopenia, which was initially defined as the loss of skeletal muscle mass that occurs with aging [13], has garnered attention as a new and promising prognostic factor for various malignancies, including HCC. For instance, skeletal muscle depletion assessed by computed tomography (CT) can predict a poor prognosis in all cancer stages [14] and in sorafenib-treated patients with HCC [15], with sarcopenia and rapid skeletal muscle wasting being associated with worse survival in those with liver cirrhosis [16,17].

In addition to skeletal muscle mass, fat mass has a notable impact on the prognosis of HCC patients. High subcutaneous fat mass (SFM) volume is involved in better survival outcomes in HCC patients treated with transcatheter intra-arterial therapy [18]. Compared to low visceral fat mass (VFM) volume, high VFM volume is advantageous for long-term survival after hepatic resection of HCC [19]. High VFM without skeletal muscle mass loss is also associated with a better prognosis in HCC patients treated with sorafenib [20]. However, high VFM is a predictor of poor survival in patients with HCC treated with tyrosine kinase inhibitors, including sorafenib [21]. Excessive VFM accumulation plays a critical role in obesity-related liver carcinogenesis [22]. In addition, it is unclear whether fat mass wasting after sorafenib introduction has effects on the outcomes of HCC patients.

The purpose of this study was thus to evaluate the effect of the changes in body composition, including skeletal muscle, SFM, and VFM after sorafenib introduction on the prognosis of patients with advanced HCC.

## 2. Results

### 2.1. Comparison of Baseline Characteristics and Laboratory Data of The Sorafenib-Treated HCC Patients with and without Sarcopenia

The baseline characteristics and laboratory data of the 61 patients (54 men and 7 women, average age 67.3 years) just before the introduction of sorafenib and the comparison of the groups with (n = 25) and without sarcopenia (n = 36) are shown in Table 1. The sarcopenia group had significantly lower values than the non-sarcopenia group regarding body composition parameters for body mass index (kg/m^2^, 20.1 vs. 23.9, *p* < 0.001), skeletal muscle index (L3SMI [cm^2^/m^2^], 37.5 vs. 48.5, *p* < 0.001), SFM index (SFMI [cm^2^/m^2^], 22.1 vs. 43.8, *p* < 0.001), and VFM index (VFMI [cm^2^/m^2^], 25.5 vs. 44.3, *p* < 0.001). The administration period of sorafenib was also significantly shorter in the sarcopenia group than in the non-sarcopenia group (days; 325 vs. 544, *p* = 0.032). The mean interval of CT examination was 127.6 days. The mean values of ∆L3SMI, ∆SFMI, and ∆VFMI during almost the same period of the scan interval (120 days) were −1.70, −0.46, and 1.02 (cm^2^/m^2^/120 days), respectively. There were no significant differences in the values of ΔL3SMI, ΔSFMI, and ΔVFMI between the two groups.

### 2.2. Impact of Sarcopenia and the Changes in Body Composition on Overall Survival in the Patients with HCC Treated with Sorafenib

The 1-, 3-, and 5-year overall survival rates were 58.9, 27.2, and 10.9%, respectively, for those with sarcopenia, and 77.3, 43.8, and 29.2% respectively, for those without sarcopenia, indicating that those with sarcopenia had a significantly poorer survival than those without sarcopenia (*p* = 0.0157, Figure 1a).

Next, we examined the impact of the rapid depletion of skeletal muscle mass and fat mass on the survival of the HCC patients treated with sorafenib. Patients within the 20th percentile cutoffs for ΔL3SMI (−5.73), ΔSFMI (−5.33), and ΔVFMI (−3.95) were classified into a rapid depletion (RD) group; those who did not meet the 20th percentile cutoffs were classified into a non-RD group. There were no significant differences in the prevalence of patients with RD of ∆L3SMI (*p* > 0.999), ∆SFMI (*p* = 0.526), and ∆VFMI (*p* = 0.099) between the sarcopenia and non-sarcopenia groups (Appendix A). We then compared the overall survival between these two groups. Regarding ΔL3SMI (*p* = 0.0101, Figure 1b) and ΔSFMI (*p* = 0.0027, Figure 1c), the RD group experienced a significantly poorer survival than non-RD group, whereas for ΔVFMI (*p* = 0.453, Figure 1d), survival was comparable between both groups. Therefore, the rapid depletion of skeletal muscle mass and, especially, SFM predicts the survival of the sorafenib-treated HCC patients. We also examined the above analyses by gender. Male patients with sarcopenia (*p* = 0.025) and RD of ∆L3SMI (*p* = 0.003) and ∆SFMI (*p* = 0.007) had poorer survival (Appendix A). Female patients with RD in ∆SFMI (*p* = 0.014) had poorer survival (Appendix A).

We then classified the enrolled patients into four groups according to the cutoffs for ΔL3SMI and ΔSFMI as described above; type I: Rapid depletion seen in neither ΔL3SMI nor ΔSFMI (*n* = 39), type II: Rapid depletion seen in ΔL3SMI alone (*n* = 10), type III: Rapid depletion seen in ΔSFMI alone (*n* = 10), and type IV: Rapid depletion seen in both ΔL3SMI and ΔSFMI (*n* = 2). The patients classified into type I had a significantly better survival than both type II (*p* = 0.007) and type III (*p* = 0.009; Figure 2).

Subsequently, we aimed to determine possible prognostic factors for the enrolled patients using the Cox proportional-hazards model. Multivariate analysis demonstrated that the presence of sarcopenia (yes vs. no; hazard ratio [HR]: 2.453, 95% confidence interval [CI]: 1.273–4.728, *p* = 0.007), ΔL3SMI (≤−5.73 vs. >−5.73; HR: 4.010, 95% CI: 1.799–8.938, *p* = <0.001), ΔSFMI (≤−5.33 vs. >−5.33; HR: 4.109, 95% CI: 1.967–8.584, *p* = <0.001), and therapeutic effect (progressive disease vs. complete response/partial response/stable disease; HR: 4.603, 95% CI: 2.188–9.683, *p* = <0.001) were independent predictors of survival for the HCC patients treated with sorafenib (Table 2).

## 3. Discussion

Sarcopenia is associated with poor outcomes in patients with HCC [14,15]. Skeletal muscle in HCC patients decreases according to worsening liver functional reserve and larger tumor size [23]. In this study, 41% of the enrolled patients were already sarcopenic when they started sorafenib treatment, suggesting that sorafenib might often be used in HCC patients with sarcopenia. The results of this study revealed that sarcopenia is an independent predictive factor of survival in HCC patients treated with sorafenib, which is consistent with the results of a previous report [15]. Because the presence of sarcopenia was associated with a shorter administration period of sorafenib, sorafenib might not have been sufficiently administered to sarcopenic patients in the present study. Moreover, rapid skeletal muscle wasting after introducing sorafenib predicts worse survival in patients with HCC. Patients with rapid depletion of skeletal muscle mass are considered to already be sarcopenic or expected to experience sarcopenia in the near future. Therefore, adequate measures for sarcopenia are required before HCC reaches an advanced stage, and even before HCC emerges. Furthermore, there is a possibility that sorafenib itself inhibits muscle protein synthesis directly [24], indicating that HCC patients treated with sorafenib are prone to sarcopenia. In order to improve the prognosis of HCC patients, it is necessary to pay more attention to skeletal muscle depletion during sorafenib treatment.

In addition to sarcopenia, patients with advanced malignancies frequently experience malnutrition and cachexia, both of which show total body weight loss, adipose tissue depression, and muscle atrophy [25,26,27]. To the best of our knowledge, this is the first study demonstrating that the rapid depletion of SFM is an independent prognostic factor in the patients with advanced HCC treated with sorafenib. Although the definition of cachexia does not necessarily require the depletion of SFM, SFM depletion is often observed in cachectic patients [26,28,29]. It has been reported that subcutaneous adipose tissue is beneficial for lipid and glucose metabolism [30]. Adipose tissue works as an energy storage and is thus able to protect cancer patients against increased energy exhaustion induced by cachexia [31]. Therefore, regarding HCC patients with rapid SFM depletion, it should be considered that they suffer from malnutrition and, in that case, it is crucial to correct the negative energy balance that led to the malnutrition. In the present study, compared to the patients without rapid depletion of both skeletal muscle mass and SFM, those with either rapid depletion of skeletal muscle mass or SFM showed a significantly worse prognosis. Loss of skeletal muscle is regarded as one of the diagnostic criteria with regard to the definitions of both cancer cachexia and sarcopenia, but SFM is not [27,32]. Therefore, considering these definitions, those with SFM alone, who have a poor prognosis, could be overlooked. Evaluation of both skeletal muscle mass and SFM throughout the period of HCC treatment is important to improve the prognosis and quality of life of the patients.

In order to manage cancer cachexia and sarcopenia adequately, it is critical to diagnose and treat these conditions before they reach a refractory stage at which interventions are ineffective [33]. Anorexia and malnutrition are significantly associated with the reduction or discontinuation of sorafenib [7], which leads to a poor prognosis in HCC. If the depletion of skeletal muscle and/or SFM is observed, effective nutritional interventions should be administered.

In the present study, rapid depletion of VFM was not associated with prognosis in HCC patients treated with sorafenib. Currently, it is unclear whether the accumulation of VFM is beneficial or harmful with respect to the prognosis of patients with HCC [19,20,21]. However, it should be considered that overnutrition and the accumulation of visceral adipose tissue are involved in liver carcinogenesis [22]. There is a significant relationship between the risk of HCC recurrence after curative treatment and various obesity-related disorders such as insulin resistance, oxidative stress, and hyperleptinemia [34,35,36,37]. Excessive accumulation of VFM, which is associated with hepatic inflammation [38], is an independent risk factor for HCC recurrence after curative treatment [39]. Therefore, maintaining the ideal body composition including both skeletal muscle mass and fat mass is required for patients with chronic liver disease.

This study has several limitations. First, it was a retrospective, single-center study, and the sample size was comparatively small. Thus, a prospective study involving a larger number of patients should be performed to validate the findings of the present study. Second, because of the retrospective design of our study, muscle strength, including grip strength and walking speed, which is usually regarded as a diagnostic criterion for sarcopenia [32], was not assessed. Third, we used arbitrary cutoff values for ΔL3SMI and ΔSFMI that were determined based on only the 20th percentiles of these variables, although these cutoff values were notably useful in screening the poor prognosis group. It is important to avoid overlooking such groups because of inappropriate cut-offs. Therefore, further long-term prospective studies including a larger number of patients are also needed to determination of the optimal cut-offs of these variables. Fourth, inter-observer reproducibility was not evaluated in this study because there was only one in our institution who could measure these parameters. However, we think it should be so excellent as intra-observer reproducibility because the areas of skeletal muscle mass and fat were almost automatically calculated by the SYNAPSE VINCENT software.

## 4. Materials and Methods

### 4.1. Patients, Treatment, and Follow-up Strategy

The patient selection in this study is shown in Figure 3. Among the 76 patients treated with sorafenib for advanced HCC in our hospital between May 2009 and December 2017, 11 were excluded because they did not take the agent for more than a month and 4 were excluded because they did not undergo abdominal CT before and after the introduction of sorafenib. The remaining 61 patients were enrolled in this study.

The objective of the sorafenib introduction was determined according to the Clinical Practice Guidelines for HCC issued by the Japan Society of Hepatology (JSH) [40]. When introducing sorafenib, all the patients in this study received explanations from their attending doctor, pharmacist, and nurse about how to take this drug, which complications (including appetite loss) could occur, and how to prevent and treat complications by using a handbook the drug manufacturer (Bayer HealthCare Pharmaceuticals, Berlin, Germany) published for patients treated with sorafenib. In particular, to prevent appetite loss, the patients were advised to eat freely and drink enough water. However, if appetite loss occurred, they were advised to feel free to consult medical stuff. When patients lost their appetite or showed signs of malnutrition such as hypoalbuminemia, branched-chain amino acids or other dietary supplement and antiemetic were prescribed as needed. Patients were thereafter followed on an outpatient basis and underwent dynamic CT, magnetic resonance imaging, or ultrasound every 3 months. Each patient’s therapeutic response was judged according to the Response Evaluation Criteria in Cancer of the Liver [41], which is an appropriate system for the assessment of the post-therapeutic response of HCC to sorafenib [42]. Overall survival was defined as the interval from the date of sorafenib introduction to the date of death or December 2018 for surviving patients. All study participants provided verbal informed consent, which was considered sufficient because this study followed an observational research design that did not require new human biological specimens. The study design, including this consent procedure, was approved by the ethics committee of the Gifu University School of Medicine (ethical protocol code: 29–26).

### 4.2. Image Analysis of Skeletal Muscle Mass and Subcutaneous and Visceral Fat Mass

Skeletal muscle mass, SFM, and VFM were measured using an enhanced CT image (Discovery CT 750 HD, Revolution CT; GE Healthcare, Milwaukee, WI, USA) that had been taken solely for the purpose of diagnosing HCC prior to introducing sorafenib in Gifu University Hospital. The CT imaging conditions in this study were as follows; the tube voltage was 120 kV, slice thickness was 5 mm, air calibration scans were performed every day, and water phantom calibration by the manufacturer was performed every 3 months. A transverse CT image at the third lumbar vertebra (L3) in the inferior direction was assessed. The muscles in the L3 region were analyzed using SYNAPSE VINCENT software (Fujifilm Medical, Tokyo, Japan), which enables specific tissue demarcation using Hounsfield unit (HU) thresholds. The muscles were quantified within a range of −29 to +150 HU [43], and tissue boundaries were manually corrected as needed. The cross-sectional areas of the muscle (cm^2^) at the L3 level computed from each image were normalized by the square of the height (m^2^) to obtain the L3 skeletal muscle index (L3SMI, cm^2^/m^2^). Sarcopenia was defined as an L3SMI value ≤38.0 cm^2^/m^2^ for women and ≤42.0 cm^2^/m^2^ for men, according to JSH guidelines for sarcopenia [32]. In the same manner, the cross-sectional areas of SFM and VFM (cm^2^) at the umbilical point were measured using a built-in function in the SYNAPSE VINCENT software. These values were then normalized by the square of the height (m^2^) to obtain the SFM index (SFMI, cm^2^/m^2^) and VFM index (VFMI, cm^2^/m^2^). One trained physician (K.I.) measured these parameters. To evaluate intra-observer reproducibility, the physician measured 30 cases of randomly selected data two times at least 3 months apart. The intra-observer reproducibilities of L3SMI, SFMI, and VFMI were excellent with intraclass correlation coefficients for all these parameters, exceeding 0.99 (Appendix A). The CT level at the umbilical point, which is most appropriate for evaluating the volume of adipose tissue [44], was used to calculate both SFMI and VFMI.

The differences in L3SMI, SFMI, and VFMI between post- and pre-sorafenib introduction were divided by CT examination interval to obtain ΔL3SMI, ΔSFMI, and ΔVFMI (cm^2^/m^2^/120 days), respectively. The outline and formula of ΔL3SMI, ΔSFMI, and ΔVFMI are shown in Figure 4.

### 4.3. Statistical Analysis

Baseline characteristics were compared using the Student’s *t*-test for continuous variables or the χ^2^ test for categorical variables. Overall survival was estimated using the Kaplan–Meier method. Differences between curves were evaluated using the log-rank test, and when there were more than three curves, Holm correction was used to counteract the problem of multiple comparisons. The Cox proportional-hazards model was used to analyze which factors, including ΔL3SMI, ΔSFMI, and ΔVFMI, affected overall survival. Statistical significance was defined as *p* < 0.05. All statistical analyses were performed using R ver. 3.3.1 (R Foundation for Statistical Computing, Vienna, Austria; http://www.R-project.org/).

## 5. Conclusions

Sarcopenia and rapid depletions of SFM and skeletal muscle mass after sorafenib introduction were associated with a poor survival in patients with advanced HCC. Adequate treatments for underlying liver diseases, especially nutritional intervention, should be conducted to avoid these unfavorable changes in body composition, which could help improve the prognosis of HCC patients treated with sorafenib.

## Figures and Tables

**Figure 1 cancers-11-01206-f001:**
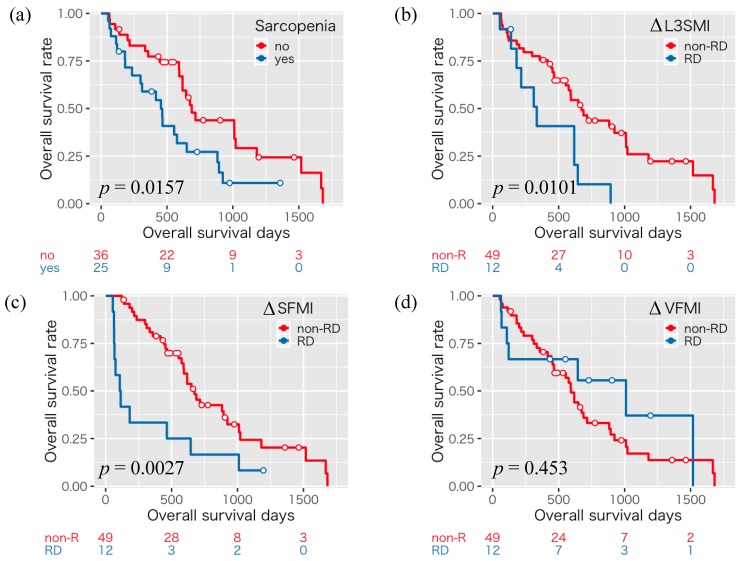
Kaplan–Meier curves for overall survival time divided into the presence or absence of sarcopenia (**a**) and rapid depletion (RD) or non-rapid depletion (non-RD) groups in ΔL3SMI (≤−5.73 and >−5.73 cm^2^/m^2^/120 days) (**b**), ΔSFMI (≤−5.33 and >−5.33 cm^2^/m^2^/120days) (**c**), and ΔVFMI (≤−3.95 and >−3.95 cm^2^/m^2^/120 days) (**d**).

**Figure 2 cancers-11-01206-f002:**
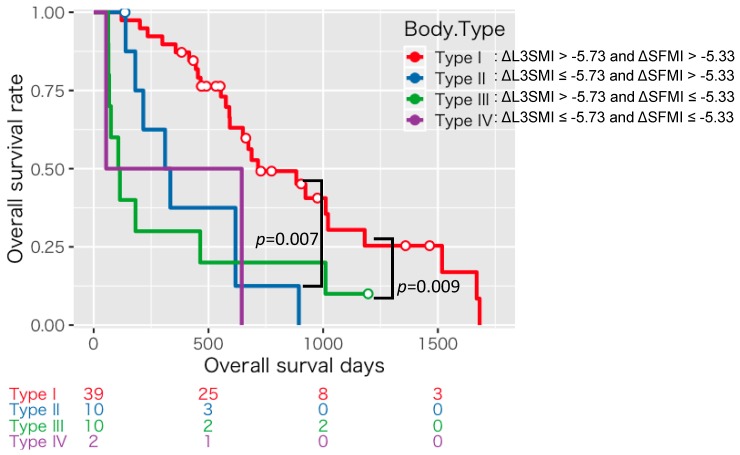
Kaplan–Meier curves for overall survival time divided into four groups according to the cutoffs for ΔL3SMI (−5.73 cm^2^/m^2^/120 days) and ΔSFMI (−5.33 cm^2^/m^2^/120 days).

**Figure 3 cancers-11-01206-f003:**
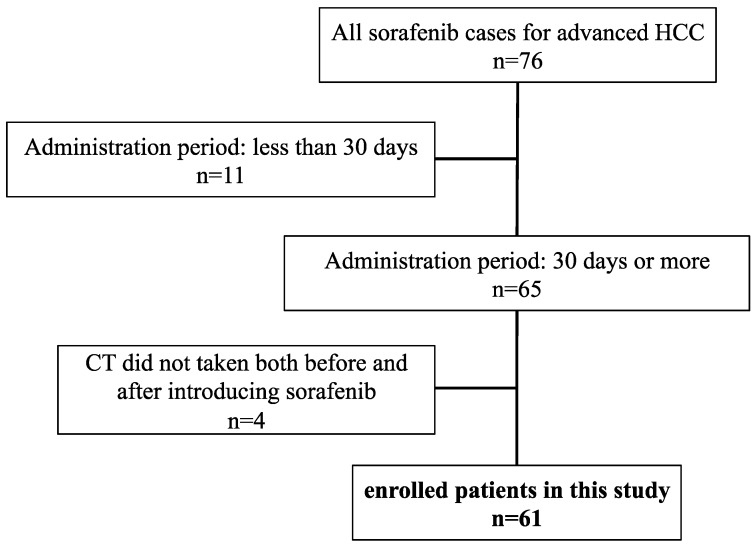
Patient flow in this study.

**Figure 4 cancers-11-01206-f004:**
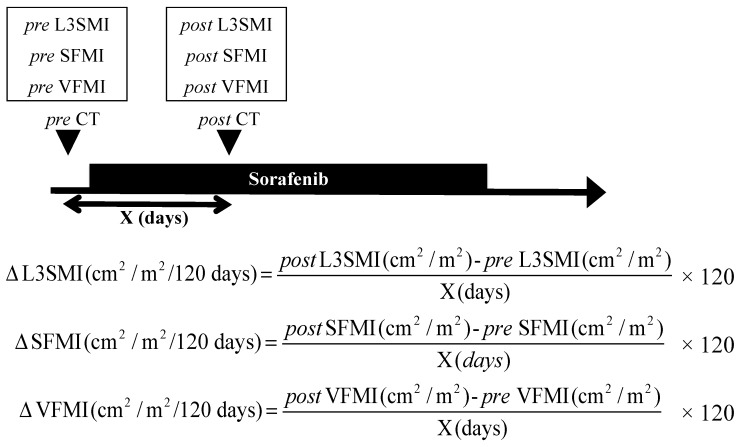
Outline and formula for ΔL3SMI, ΔSFMI, and ΔVFMI. L3 skeletal muscle index (L3SMI) was the cross-sectional area of the muscle (cm^2^) at the L3 level of the computed tomography (CT) image normalized by the square of the height (m^2^). Subcutaneous fat mass index (SFMI) and visceral fat mass index (VFMI) were the cross-sectional areas of the subcutaneous and visceral fat (cm^2^), respectively, at the umbilical point normalized by the square of the height (m^2^).

**Table 1 cancers-11-01206-t001:** Baseline demographic and clinical characteristics of patients with and without sarcopenia.

Variables		All Cases (n = 61)	Non-Sarcopenia (n = 36)	Sarcopenia (n = 25)	*p*-Value
Sex (male/female)		54/7	32/4	22/3	>0.999
Age (years)		67.3 ± 11.5	67.1 ± 11.7	67.7 ± 11.3	0.826
Etiology (B/C/others)		14/28/19	6/17/13	8/11/6	0.504
Child-Pugh score (5/6)		43/18	27/9	16/9	0.401
Stage (III/IVA/IVB)		20/13/28	11/10/15	9/3/13	0.348
Combination therapy (yes/no)		42/19	24/12	18/7	0.781
BMI (kg/m^2^)		22.3 ± 3.0	23.9 ± 2.4	20.1 ± 2.4	<0.001
L3SMI (cm^2^/m^2^)	All	44.0 ± 7.7	48.5 ± 6.1	37.5 ± 4.6	<0.001
	Male	44.5 ± 7.7	48.9 ± 6.2	38.2 ± 4.4	<0.001
	Female	39.9 ± 7.4	45.2 ± 4.3	32.7 ± 2.3	0.006
SFMI (cm^2^/m^2^)	All	34.9 ± 22.0	43.8 ± 21.9	22.1 ± 14.8	<0.001
	Male	31.5 ± 18.9	39.5 ± 17.8	19.9 ± 13.7	<0.001
	Female	61.2 ± 28.1	78.0 ± 24.0	38.8 ± 4.6	0.054
VFMI (cm^2^/m^2^)	All	36.6 ± 21.0	44.3 ± 20.5	25.5 ± 16.4	<0.001
	Male	31.5 ± 18.4	44.9 ± 20.7	26.1 ± 17.2	<0.001
	Female	37.2 ± 21.4	39.4 ± 20.9	21.0 ± 8.2	0.217
∆L3SMI (cm^2^/m^2^/120 days)		−1.70 ± 7.96	−2.14 ± 4.82	−1.07 ± 11.13	0.609
∆SFMI (cm^2^/m^2^/120 days)		−0.46 ± 11.34	0.35 ± 13.17	−1.62 ± 8.12	0.509
∆VFMI (cm^2^/m^2^/120 days)		1.02 ± 12.11	0.72 ± 13.96	1.44 ± 9.08	0.821
CT examination interval (days)		127.6 ± 89.5	127.0 ± 62.9	128.5 ± 119.4	0.949
Administration period of sorafenib (days)		455 ± 396	544 ± 448	325 ± 264	0.032
Therapeutic effect (CR/PR/SD/PD)		3/34/5/19	2/18/2/14	1/16/3/5	0.368

Values are presented as mean ± standard deviation. Sarcopenia was defined as an L3SMI value of ≤38.0 cm^2^/m^2^ for women and ≤42.0 cm^2^/m^2^ for men. B, hepatitis B virus; C, hepatitis C virus; BMI, body mass index; L3SMI, third lumbar vertebra skeletal muscle index; SFMI, subcutaneous fat mass index; VFMI, visceral fat mass index; CR, complete response; PR, partial response; SD, stable disease; PD, progressive disease.

**Table 2 cancers-11-01206-t002:** Univariate and multivariate analyses of possible prognostic factors in patients with hepatocellular carcinoma treated sorafenib, according to the Cox proportional-hazards model.

Variables	Univariate Analysis		Multivariate Analysis	
	HR (95% CI)	*p*-Value	HR (95% CI)	*p*-Value
Sex (male vs. female)	0.872 (0.365–2.082)	0.758		
Age (years)	0.980 (0.956–1.006)	0.127		
Child-Pugh score (6 vs. 5)	1.630 (0.850–3.154)	0.141		
Stage (III vs. IV)	1.009 (0.526–1.935)	0.979		
Combination therapy (yes vs. no)	1.007 (0.508–1.995)	0.984		
BMI (kg/m^2^)	0.954 (0.846–1.075)	0.440		
Sarcopenia (yes vs. no)	2.124 (1.137–3.967)	0.018	2.453 (1.273–4.728)	0.007
∆L3SMI (≤−5.73 vs. >−5.73)	2.560 (1.218–5.377)	0.013	4.010 (1.799–8.938)	<0.001
∆SFMI (≤−5.33 vs. >−5.33)	2.771 (1.382–5.559)	0.004	4.109 (1.967–8.584)	<0.001
∆VFMI (≤−3.95 vs. >−3.95)	0.733 (0.324–1.657)	0.456		
Therapeutic effect (PD vs. CR/PR/SD)	3.049 (1.563–5.949)	0.001	4.603 (2.188–9.683)	<0.001

HR, hazard ratio; CI, confidence interval; BMI, body mass index; ∆L3SMI, change in third lumbar vertebra skeletal muscle index; ∆SFMI, change in subcutaneous fat mass index; ∆VFMI, change in visceral fat mass index; CR, complete response; PR, partial response; SD, stable disease; PD, progressive disease.

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
