# Peer review of "Rapid Depletions of Subcutaneous Fat Mass and Skeletal Muscle Mass Predict Worse Survival in Patients with Hepatocellular Carcinoma Treated with Sorafenib"

_cancers, 2019, doi:10.3390/cancers11081206_

Round 1

Reviewer 1 Report

This article suggests that both reduced muscle mass at baseline and loss of muscle and fat over time are significant independent predictors of survival in patients with HCC treated with sorafenib.

Loss of muscle and fat overtime are reported in this manuscript. I cannot find anywhere in the document how long is the time interval, in days, between the first scan and the last scan for each patient. This should be reported. The data are expressed as %change per month; This is rather odd because the scans are certainly not one month apart. I realize this has been done in order to account for variation in the scan interval, However if the mean scan interval is 100 days then it would be more sensible to express the results as percentage change / 100 days. Expressing the data over a longer period of time will not change the results however what is strange about expressing the change over a very short time of 1 month is that the values are small enough to be the same as the intra- or inter-observer variation in the measurements in question.

Table 1 clarification :The criteria for the definition of Sarcopenia should be included in the legend to table one. The values for L3 SMI, SFMI and VFMI are sex dependent; therefore data for men and women for these parameters should be shown separately in table one. It is not clear what “administration period“ refers to in table 1.

It’s a conventional in these types of studies to give more detail about the CT image analysis. Where are the CT’s taken all on the same instrument? How was the instrument calibrated? What was the tube voltage? What was that slice thickness? Are the images contrast enhanced or unenhanced? Who use this software to quantify the tissue areas and what was their training? Were they radiologist? Was there more than one observer? What is the individual intra-observer coefficient of variation for the measures? If there was more than one observer what was the inter-observer coefficient of variation? What was the mean time in days between scans?

The umbilicus is not a bony landmark and normally this would not be considered acceptable for land marking images. This analysis is usually done at a specified lumber vertebra.

In the results section I would like to know if there are any sex differences. This comment is raised because I visceral fat is a male characteristic and high subcutaneous fat is a female characteristic.

The discussion requires major revision….

In sections starting on page 6, outline 143 it is extremely confusing that the discussion is talking about malnutrition as well as cachexia and sarcopenia. I believe the main point of this paragraph is to suggest to the reader that the CT measurements add value to conventional diagnostic criteria, however we have three different diagnoses being discussed, a diagnosis of cachexia, a diagnosis of sarcopenia and a diagnosis of malnutrition. Raising all these terms without giving their explicit definitions and relationship to one another is just confusing.

This section of discussion from line 159 to 168, Is not supported by the data presented in this paper. I think this section should be deleted as the results of the study provide no basis to suggest that amino acids or exercise should be recommended.

Line 145-146 refs 25 and 26 are cited here, however the topics of these references do not correspond to the statements made.

Author Response

Reviewer #1

This article suggests that both reduced muscle mass at baseline and loss of muscle and fat over time are significant independent predictors of survival in patients with HCC treated with sorafenib.

Loss of muscle and fat overtime are reported in this manuscript. I cannot find anywhere in the document how long is the time interval, in days, between the first scan and the last scan for each patient. This should be reported. The data are expressed as %change per month; This is rather odd because the scans are certainly not one month apart. I realize this has been done in order to account for variation in the scan interval, However if the mean scan interval is 100 days then it would be more sensible to express the results as percentage change / 100 days. Expressing the data over a longer period of time will not change the results however what is strange about expressing the change over a very short time of 1 month is that the values are small enough to be the same as the intra- or inter-observer variation in the measurements in question.

In the present study, the mean interval of CT examination was 127.6 days (lines 81 to 82 and new Table 1). The mean values of ∆L3SMI, ∆SFMI, and ∆VFMI during the almost same period of the interval (120 days) were -1.70, -0.46, and 1.02 (cm2/m2/120 days), respectively (line 19, lines 24-25, lines 82 to 83, new Tables 1 and 2, and new Figures 1, 2 and 4). We thank your valuable comment that improves the quality of our manuscript.

Table 1 clarification: The criteria for the definition of Sarcopenia should be included in the legend to table one. The values for L3 SMI, SFMI and VFMI are sex dependent; therefore data for men and women for these parameters should be shown separately in table one. It is not clear what “administration period“ refers to in table

According to this indication, we added new data relating to sex differences and revised the new Table 1

It’s a conventional in these types of studies to give more detail about the CT image analysis. Where are the CT’s taken all on the same instrument? How was the instrument calibrated? What was the tube voltage? What was that slice thickness? Are the images contrast enhanced or unenhanced? Who use this software to quantify the tissue areas and what was their training? Were they radiologist? Was there more than one observer? What is the individual intra-observer coefficient of variation for the measures? If there was more than one observer what was the inter-observer coefficient of variation? What was the mean time in days between scans?

According to this suggestion, we described more detail about the CT image analysis in the Materials and Methods section (lines 232-237). In this study, one trained physician (K.I.) measured the parameters. To evaluate observer reproducibility, he measured 30 cases of randomly selected data at two times at least three months apart. The intra-observer reproducibilities of L3SMI, SFMI, and VFMI were excellent with intraclass correlation coefficients for all these parameters, exceeding 0.99 (lines 248 to 252 and Supplementary Figure 3). Inter-observer reproducibility was not evaluated in this study. The areas of skeletal muscle mass and fat were almost automatically calculated by SYNAPSE VINCENT software; however, the loss of inter-observer reproducibility might be involved in the occurrence of a bias. We understand this is one of the limitations of the study and mentioned this point in the Discussion section (lines 196-200). We thank your important comment again.

The umbilicus is not a bony landmark and normally this would not be considered acceptable for land marking images. This analysis is usually done at a specified lumber vertebra.

In the present study, CT level at umbilical point was used to calculate both SFMI and VFMI because this point is most appropriate to evaluate the volume of adipose tissue as described previously [new Ref. 44]. We rewrote the Materials and Methods section (lines 252-253) by citing new reference #45.

In the results section I would like to know if there are any sex differences. This comment is raised because I visceral fat is a male characteristic and high subcutaneous fat is a female characteristic.

In answer to your request, we analyzed the impact of sarcopenia and the changes in body composition on the overall survival by gender (new supplementary Figures 1 and 2). Male patients with sarcopenia (p = 0.025) and RD of ∆L3SMI (p = 0.003) and ∆SFMI (p= 0.007) had poorer survival. Female patients with RD in ∆SFMI (p = 0.014) had poorer survival. These results were described in the Results section (lines 108-110).

In sections starting on page 6, outline 143 it is extremely confusing that the discussion is talking about malnutrition as well as cachexia and sarcopenia. I believe the main point of this paragraph is to suggest to the reader that the CT measurements add value to conventional diagnostic criteria, however we have three different diagnoses being discussed, a diagnosis of cachexia, a diagnosis of sarcopenia and a diagnosis of malnutrition. Raising all these terms without giving their explicit definitions and relationship to one another is just confusing.

It is well known that, in addition to sarcopenia, patients with advanced malignancies frequently suffer from malnutrition and cachexia, both of which show total body weight loss, adipose tissue depression, and muscle atrophy [new Ref. 25-27]. Adipose tissue works as an energy storage and is thus able to protect cancer patients against increased energy exhaustion induced by cachexia [31]. Therefore, we discussed the impact of adipose tissue depression with focusing on malnutrition and cachexia. We clarified this point in the revised text (lines 154 to 156). We appreciate your important opinion.

This section of discussion from line 159 to 168, Is not supported by the data presented in this paper. I think this section should be deleted as the results of the study provide no basis to suggest that amino acids or exercise should be recommended.

According to this comment, we deleted the suggested sentence and revised the text (lines 175 to 176). We thank your appropriate suggestion again.

Line 145-146 refs 25 and 26 are cited here, however the topics of these references do not correspond to the statements made.

According to this comment, we modified appropriate expressions to correspond to the content of the references and add a new reference (lines 158 to 159, and new reference 28).

Reviewer 2 Report

Overall a good presentation of data

Numbers are small and the 20 percentile is an artificial new parameter to show a statistically significant difference between groups for subcutaneous fat composition

There is no mention whether any of the patients received dietary advice and / or nutritional supplementation. It would be useful to have a description of what their practise is regarding nutritional support of patients

How many of the 12 RD patients  were also in the sarcopenia group?

Author Response

Reviewer #2

Overall a good presentation of data

Numbers are small and the 20 percentile is an artificial new parameter to show a statistically significant difference between groups for subcutaneous fat composition.

We really understand that your comments are critical limitations of this study. As you suggested, the sample size of the study was comparatively small and, therefore, a prospective study involving a larger number of patients should be performed to validate the findings of the present study (lines 188 to 189). The 20 percentile might be an artificial new parameter although it shows a statistically significant difference between the groups. We consider it is important to avoid overlooking such group because of inappropriate cut-offs. Therefore, further long-term prospective studies including a larger number of patients are also needed to determination of the optimal cut-offs of these variables. We emphasized these points in the revised text (lines 194 to 196). We deeply appreciate your important suggestion.

There is no mention whether any of the patients received dietary advice and / or nutritional supplementation. It would be useful to have a description of what their practice is regarding nutritional support of patients

According to this suggestion, we described the instructions of daily life and nutrition when taking sorafenib (lines 209 to 218). We believe this added description is helpful for the readers. We thank your important suggestion again.

How many of the 12 RD patients were also in the sarcopenia group?

There were not significant differences in the prevalence of patients with RD of ∆L3SMI (p= 1.000), ∆SFMI (p= 0.526), and ∆VFMI (p= 0.099) between sarcopenia and non-sarcopenia groups (lines 100-103). The details of the results were shown in Supplementary

Round 2

Reviewer 1 Report

Thank you for very thorough revisions to this document. I find these changes to be satisfactory